

# A thinner-than-present West Antarctic Ice Sheet in the southern Weddell Sea Embayment during the Holocene.

David Small[1]*, Réka-H. Fülöp[2,3], Rachel K. Smedley[4], Tom Lees[1], Stephan Trabucatti[1,5], Derek Fabel[6], Maria Miguens-Rodriguez[6], Andrew M. Smith[3], Grant V. Boeckmann[7].

[1] Department of Geography, Durham University, Durham, UK

[2] School of Earth, Atmospheric and Life Sciences, University of Wollongong, NSW, Australia

[3] Australian Nuclear Science and Technology Organisation (ANSTO), Lucas Heights, NSW, Australia

[4] Department of Geography and Planning, University of Liverpool, Liverpool, UK

[5] 3D Drilling, East Grinstead, UK

[6] Scottish Universities Environmental Research Center (SUERC), University of Glasgow, East Kilbride, UK

[7] Niels Bohr Institute, University of Copenhagen, Copenhagen, Denmark

Correspondence to: David Small (david.p.small@durham.ac.uk)

**Abstract.** Making accurate measurements and predictions of the West Antarctic Ice Sheet's (WAIS) contribution to present and future sea-level rise fundamentally depends on knowing its trajectory over the last few thousand years. We present new in situ $^{14}$C concentrations from subglacial bedrock cores collected from the southern Weddell Sea sector of the WAIS. Critically, these concentrations are above levels that can be produced under present-day ice thicknesses at the core sites. The cosmogenic nuclide inventories provide clear evidence for the ice sheet being thinner-than present at some point during the Holocene following initial thinning from its Last Glacial Maximum configuration. Forward modelling of nuclide concentrations indicates that the nuclide depth-profiles within our cores are best explained by a 500 - 4000 year period of (near) total exposure that has occurred since 6-4 ka. We suggest that thinning at our core sites is most likely to reflect a regional, dynamic response to grounding-line retreat rather than a localised change in ice-surface elevation. Our data are the first direct geological evidence for a thinner-than-present WAIS in the Weddell Sea sector and are consistent with Holocene retreat that culminated inboard of present-day limits. Glacio-isostatic adjustment has been inferred as a driving mechanism, causing re-grounding of floating ice and increased buttressing allowing the grounding line to stabilise and readvance. These data allow dynamic retreat-readvance behaviour of this nature to be tested in ice-sheet models, improving predictions of future sea-level rise in this critical sector of West Antarctica.



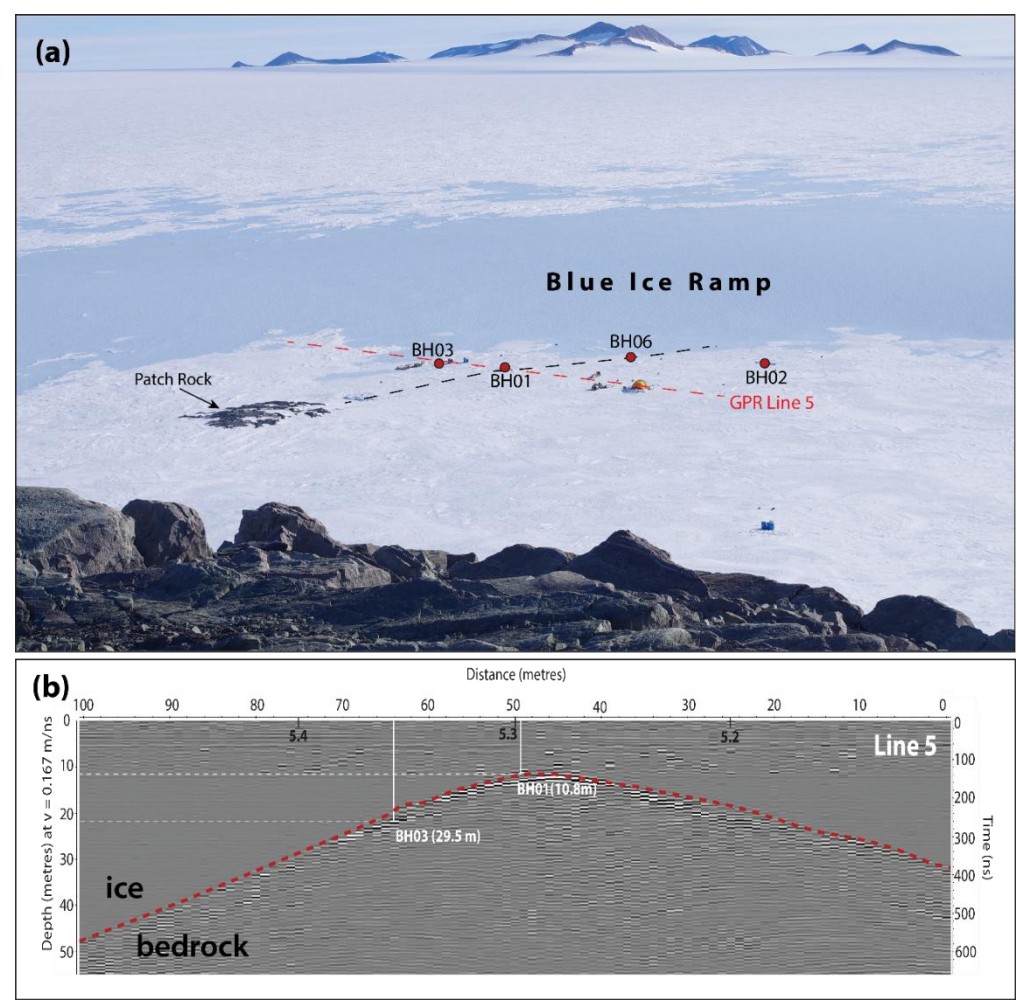



## 1 Introduction

Constraining the recent (i.e. Holocene) trajectory of the West Antarctic Ice Sheet (WAIS) is critical for making accurate
measurements of current ice loss and for making predictions of its future evolution.  In the first case, glacio-isostatic adjustment
(GIA) is a key correction for current estimates of mass loss by satellite gravimetry (e.g. Caron and Ivins, 2020). Depending on
the recent ice-sheet history the magnitude and/or sign of the GIA signal (i.e. uplift vs. subsidence) can be different (Bradley et
al., 2015; Gomez et al., 2018; King et al., 2022). With sparse observations recent ice-loading histories are required for the GIA

models that are used to correct satellite gravimetric estimates of current ice mass loss. In the second case numerical ice-sheet
models used to make predictions of future sea-level change are tested (and tuned) against their ability to accurately reproduce
known scenarios of past ice-sheet change (DeConto and Pollard, 2016; Albrecht et al., 2020; Pittard et al., 2022).

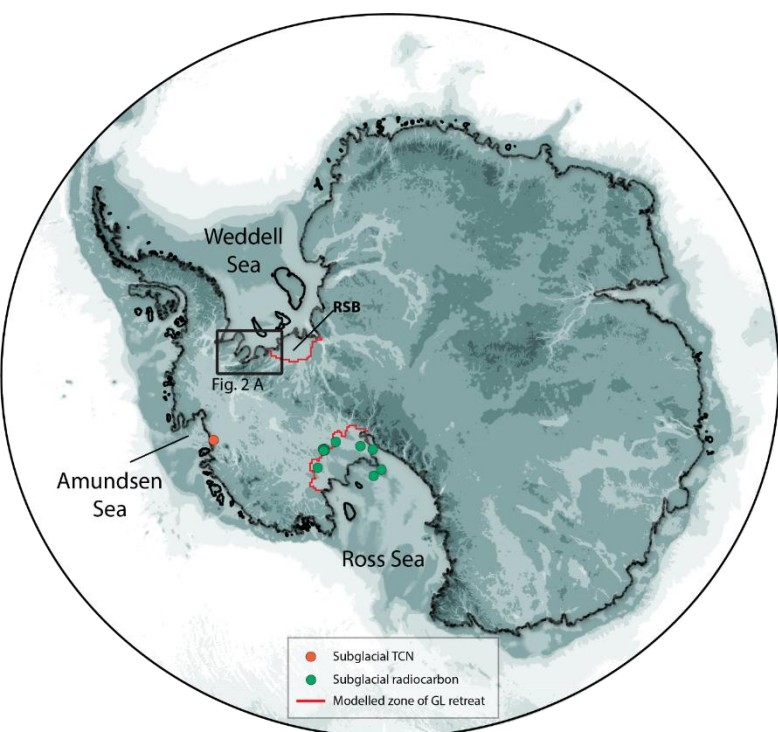

**Figure 1. Overview map of Antarctica showing locations where geological evidence of Holoence retreat inboard of the present day
grounding line has been recovered. Modelled zones of retreat in the Ross and Weddell Sea embayments are taken from Kingslake
et al. (2018), other models show similar patterns. Black box shows location of Fig. 2a. RSB: Robin subglacial basin. Base image is
BEDMAP3 subglacial topography (Pritchard et al., 2025).**




There is increasing evidence that parts of the WAIS retreated inboard of their pre-industrial limits before readvancing to these limits in the mid-late Holocene (for a review see Johnson et al., 2022 and Jones et al., 2022). Direct geological evidence for retreat-readvance, in the form of measurable amounts of radiocarbon in subglacial sediments and inventories of cosmogenic

nuclides in subglacial bedrock (Fig. 1), has been observed in both the Ross and Amundsen Sea sectors of the WAIS (Kingslake et al., 2018; Venturelli et al., 2023; Balco et al., 2023). Several numerical models also retrodict major retreat-readvance in the southern Weddell Sea Embayment (WSE) of West Antarctica (Kingslake et al., 2018; Albrecht et al., 2020 Pittard et al., 2022) where ice is only lightly grounded over the deep Robin Subglacial Basin (Ross et al., 2012). Indirect evidence, such as low present-day uplift rates (Bradley et al., 2015) and (sub-) surface morphology of ice rises (Siegert et al., 2013; Kingslake et al.,

2018; Brisbourne et al., 2019), is consistent with retreat-readvance but could also be explained by major flow re-organisation without grounding line retreat (Siegert et al., 2019). Consequently, the Holocene trajectory of the WAIS in the WSE is yet to be unequivocally constrained. This is of critical importance as the Weddell Sea is a key sector for both the WAIS and EAIS, draining ~20% of their total grounded ice area with a SLE of >12 m (Rignot et al., 2019), and a key locale for global oceanic circulation with up to ~50% of Antarctic Bottom Water formation occurring there (Zhou et al., 2023).


Here we present the first analyses from subglacial bedrock cores collected from the southern WSE to directly test for Holocene retreat-readvance of the WAIS in this sector. The basic premise (cf. Balco et al., 2023) of our approach is that dynamic thinning associated with grounding line retreat would expose bedrock that is presently buried by several tens of metres of glacier ice, at locations upstream from the present-day grounding line. Any Holocene thinning would result in significantly increased

production rates of cosmogenic nuclides and potential exposure of bedrock surfaces to sunlight. In situ [14]C (produced in rocks) has a half-life of 5730 years which means that any [14]C present at the onset of the Last Glacial Maximum (LGM) would have been removed by radioactive decay during complete shielding by a thick LGM ice sheet. Thus, measurable [14]C in subglacial bedrock requires production to have occurred following thinning of the LGM ice sheet (i.e. during the Holocene). Complementing this, luminescence measurements in rock surfaces are sensitive to exposure to light (Sohbati et al.,2012), thus

measurements can constrain ice thinning beyond present. We present results of in situ [14]C and luminescence analyses from four bedrock cores collected during the 2022-2023 Austral summer in the Enterprise Hills, a constituent massif of the Heritage Range within the Ellsworth Mountains, southern Weddell Sea Embayment.



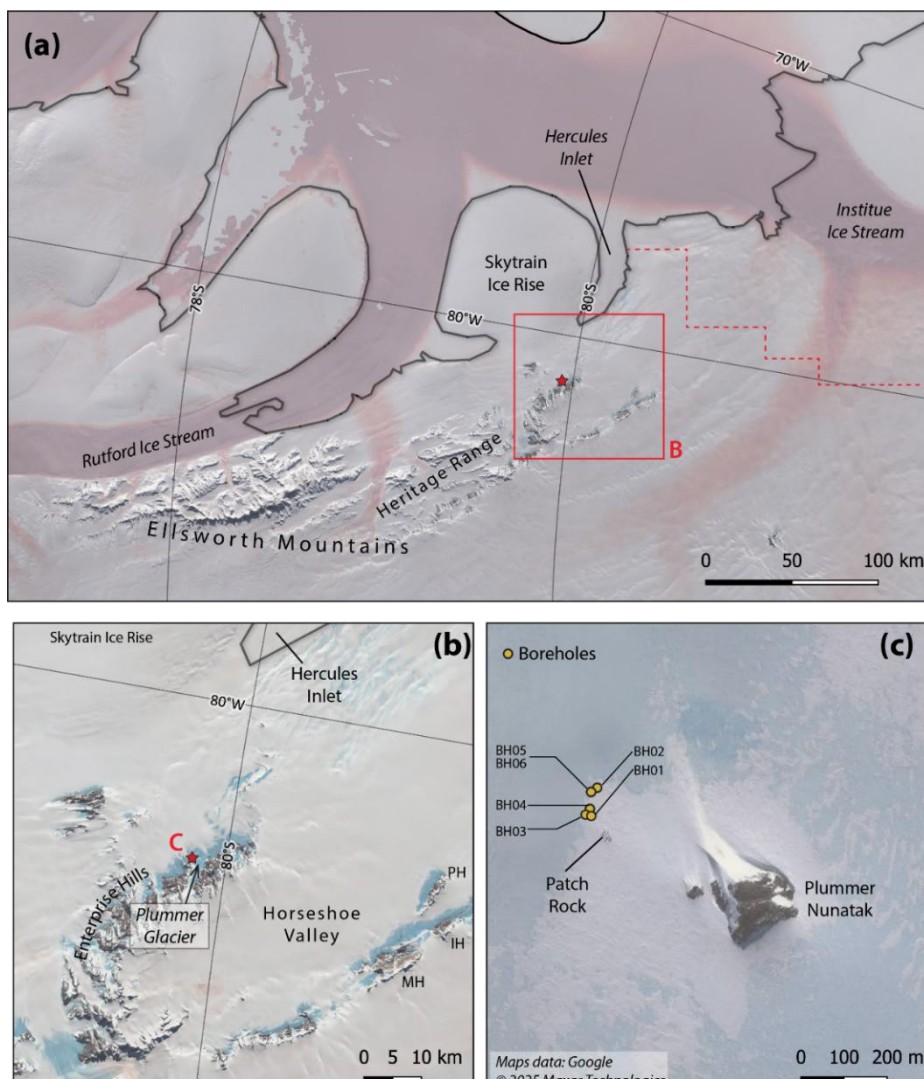

**Figure 2. a) Map of the southern WSE showing major ice streams and geographical features mentioned in the text. The dashed line is zone of modelled retreat (Kingslake et al., 2018). Red star is location of our drill site. b) Location map of Horseshoe Valley and Enterprise Hills. Location of Plummer glacier is shown. Red star is location of our drill site. MH/PH/IH = Marble/Patriot/Independence Hills where cosmogenic nuclide data evidences post LGM thinning. c) Google Earth image (© Google 2025) of Plummer Nunatak. Patch Rock is visible and locations of boreholes shown. Figure generated in Quantarctica version 3 (Matsuoka et al., 2021).**

## 2. Site description and regional ice-sheet history

The Enterprise Hill are located within the Heritage Range, ~30 km from the present-day grounding line at Hercules Inlet (Figs. 2a and 2b). While further rock outcrops closer to the grounding line, available satellite imagery suggests extensive snow/firn at these sites which prevents deployment of the Winke Drill as modified at Durham University (cf. Boeckmann et al., 2021;



Braddock et al., 2025). The Enterprise Hills delimit the northern flank of Horseshoe Valley within which ice flows in a general north-westerly direction towards the grounding line. Several glaciers breach the northern bounding escarpment of Horseshoe

Valley and merge with extensive areas of blue ice found on the leeward side of the Enterprise Hills (Fig. 2b). At the mouth of one such glacier (Plummer Glacier) is a small nunatak, informally named Plummer Nunatak, that reaches ~60 m above present-day ice. Plummer Nunatak is flanked by a small outlier (~10 m high) on its northern side. A further ~250 m from this outlier is a small patch of exposed bedrock (Fig. 2c). This outcrop (informally referred to as Patch Rock) is surrounded by a small snow field that, where probed was always < 2m thick and directly/unconformably overlies blue ice (i.e. there is no snow-firn-

ice transition). The Enterprise Hills are composed of quartzose sandstone (sometimes referred to as quartzites) of the Crashsite Formation (Spörli, 1992) and thus contain quartz suitable for cosmogenic nuclide analyses.

Following the LGM the WAIS in the region of the southern Heritage Range thinned rapidly by c. 400 m during the mid-Holocene with ice reaching present day elevations by c. 6-4 ka as evidenced by exposure age data from the Marble, Patriot,

and Independence Hills (Bentley et al., 2010; Hein et al., 2016). Additional evidence from an ice core at Skytrain Ice Rise also suggests rapid surface lowering of c.450 m at 8 ka (Griemann et al., 2024). Given the consistency between these records, it seems reasonable to infer a similar and consistent ice-sheet history across the Heritage Range, including at our site in the Enterprise Hills. Specifically, this history is inferred to be a significantly (c. 400 m) thicker ice sheet at the LGM which subsequently thinned rapidly during the Holocene to reach present day ice levels by c. 6-4 ka. The two alternate scenarios for

the subsequent ice history are 1) ice remained at that level for the remainder of the Holocene, or 2) ice thinned below present post 6-4 ka (cf. Johnson et al., 2022; Jones et al., 2022; Balco et al., 2023).

## 3. Methods

### 3.1 Subglacial bedrock sampling

To aid site selection we used a PulseEkko Pro Ground Penetrating Radar (GPR) system equipped with 100 MHz antennas,

arranged in a perpendicular broadside configuration. The antennas, transmitter, and receiver were attached to a plastic sledge and hand-towed across the survey area at ~1 m s⁻¹. Due to equipment failure, we could not co-record locational information, so survey lines were marked out by flags at 25 m intervals to allow the data to be rubber-banded in post-processing (See supplemental). Following initial survey immediately prior to commencing drilling activities, we identified suitably shallow bedrock extending north from a small exposure of bedrock close to the main outcrop of Plummer Nunatak (Fig. 3). The ridge

crest maintained a consistent depth beneath the ice surface (c. 7 – 10 m) before it passed below a steep ramp of blue ice. As a result, and to obtain cores from a range of depths, we decided to drill at two locations offset from the main ridge crest (BH02 and BH03). This introduced additional uncertainty in our estimates of ice depths due to off-nadir reflectors and restricted time for post-processing. That said, as the depths involved were comfortably within range of the Winkie Drill this was not considered to be a critical issue.






We used a modified Winkie Drill to collect subglacial bedrock samples from depths up to 29.5 m below present-day ice levels. Our drill set-up is similar to the US Ice Drilling Program Winkie Drill deployed to the Ohio Range in 2016/17 (Boeckmann et al., 2021; Braddock et al., 2025) being powered by a 2-stroke internal combustion engine rather than an electric powerhead. To make access holes we used modified Kovacs ice augers attached to the Winkie Drill transmission. Once bedrock was

reached, we replaced the augers with an AW34 drill rod string and cored bedrock using an IAWS core barrel using D60 drill fluid in a forward fluid circulation configuration. As all our boreholes were created directly into blue ice there was no requirement to case the hole.

We drilled a total of six access holes at depths of 7.0 – 29.5 m below present-day ice (Table 1). We successfully recovered

bedrock cores from four of these boreholes, BH01, BH02, BH03, and BH06 (Table 1 and Fig. S2 in supplement). All cores were composed of medium-fine grained quartz arenites of the Crashsite formation. In BH04 we were unable to establish fluid circulation, presumably due to fluid escaping in through a fracture in the bedrock.  At BH05 we hit bedrock at an unexpectedly shallow depth (7 m) and decided against sampling. At BH01 we recovered 10 cm of bedrock in our first core run but during the second core run the core barrel became irretrievably stuck due to inefficient clearance of chippings.


| Borehole | Latitude (DD) | Longitude (DD) | Ice surface elevation (m) | Borehole depth (m) | Bedrock surface elevation (m) | Core recovery (cm) |
|---|---|---|---|---|---|---|
| BH01 | -79.9439 | -81.4314 | 509 | 10.8 | 498.2 | 10 |
| BH02 | -79.9440 | -81.4288 | 507 | 24.1 | 482.9 | 47 |
| BH03 | -79.9439 | -81.4322 | 510 | 29.5 | 480.5 | 26 |
| BH04 | -79.9440 | -81.4323 | 509 | 8.5 | 500.5 | N/A |
| BH05 | -79.9439 | -81.4294 | 508 | 7.0 | 501.0 | N/A |
| BH06 | -79.9439 | -81.4294 | 509 | 9.5 | 499.5 | 5 |

**Table 1. Borehole locations, elevations, and core recovery.**





**Figure 3. A) Patch Rock viewed from the top of Plummer Nunatak. The locations of the bedrock cores are shown. The black dashed line denotes the approximate trace of the subglacial ridge as surveyed before it passes underneath the blue ice ramp. Vertical relief of the ramp is c.10 m. GPR Line 5 is shown as a red dashed line in A). B) Interpreted radargram of GPR Line 5 showing locations of cores BH01 and BH03 along with measured depths to bedrock (n.b. for BH03 this is greater than depth extrapolated from two-way-travel time). The numbers 5.2 etc denote locations of flags used as fiducial markers. GPR Lines 1-4 were located at 10 m intervals in increasing proximity to Patch Rock (see supplemental).**



## 3.2 Luminescence measurements of BH01

BH01 was recovered in the dark to facilitate analysis using rock surface luminescence (Sohbati et al., 2011). The sample was
prepared at the University of Liverpool luminescence laboratory under subdued-lighting conditions to prevent sunlight contamination of the luminescence signal. Due to the irregular core-top morphology, a single luminescence-sample core was recovered. The core was sliced to a thickness of ~0.7 mm and slices mounted in stainless steel cups for luminescence measurements. Luminescence measurements followed standard protocols employed at the University of Liverpool luminescence laboratory (see supplemental). Luminescence depth profiles were determined by measuring the natural signal
($L_n$) normalised using the signal measured in response to a test-dose ($T_n$), termed the $L_n/T_n$ signal. Dose-recovery experiments were consistent with a suitable analysis protocol (see extended discussion in supplemental).

## 3.3 In situ cosmogenic $^{14}$C analyses

Of the remaining cores, BH02 and BH03 were sliced at 5 cm intervals while BH06 was processed as a single sample. A total of ten samples were sent to the ANSTO-University of Wollongong in situ $^{14}$C extraction laboratory (Fülöp et al., 2019) for
carbon extraction. The $^{14}$C extraction scheme at ANSTO exploits the high temperature phase transformation of quartz to cristobalite to quantitatively extract the carbon as $CO_2$. As part of this process samples are pre-heated to 600°C to remove meteoric $^{14}$C. Impurities of chlorite and/or glauconite produced reactions in two of our samples (BH02 0-5 cm and BH02 10-15 cm) presumably due to presence of Cl and/or K. As a result two sample tubes cracked during in vacuo cleaning. Following recovery of these samples from the coil-heaters and subsequent carbon extraction distinct discolouration of the residual
material suggested contamination by the coil-heater itself and the initial samples were disregarded (Supplemental Fig. S10). For one sample (BH02 0-5 cm; OZ12602) additional material was recovered from the fine (75-125 µm) fraction which was subsequently extracted. All other extractions were carried out on the 125-500 µm fraction. We thus present results from a total of nine samples (Table 2). Following extraction $CO_2$ gas is converted to graphite targets and analysed using ANSTO's ANTARES 10MV tandem accelerator (Fink et al., 2004; Smith et al., 2010). Samples are normalised to small OX-II standards,
corrected for a global graphitisation blank by 1/m analysis and data subsequently reduced according to Hippe and Lifton (2014).

## 3.4 Forward modelling of in situ $^{14}$C concentrations

To explore scenarios of ice thickness change that are consistent with the measured concentrations we use a simple forward model for cosmogenic nuclide production under 10000 random three-stage ice histories (Fig. 4). Stage 1 has a random start
time ($T_1 = 10$ - 6ka) and represents the early Holocene when the regional ice-sheet history mandates that our site was covered by an ice that was thicker-than-present. During Stage 1 ice thickness ($H_{s1}$) is constrained to be no thinner than 50 m to allow for some production by deeply penetrating muons to assess sensitivity of the model to initial ice thickness. Stage 2 ($T_2 - T_3$) is a period of thinner-than-present ice that begins at a random time after $T_1$. In all boreholes we observed a clean ice-rock



transition at the bed. Balco et al. (2023) used the presence of a dirty ice layer at the base of their boreholes to infer that ice had

not thinned beyond their core top elevations during the Holocene. The absence of such a layer at our sites would, by the same

logic, imply that ice *could* have thinned below our core top elevations but does not mandate that it did so. Consequently, during

Stage 2 ice thickness ($H_{s2}$) is allowed to randomly vary between 0 m (i.e. total exposure of our cores) and its present-day

thickness with 10% of model iterations set to zero thickness. This allows our model to explore what duration of complete

exposure is consistent with our measured concentration and what thickness of residual ice cover is possible if the assumption

of complete exposure is incorrect. Finally, Stage 3 ($T_3$ - $T_4$) represents the current period of burial by ice with its present-day

thickness ($H_{S3}$). The start-time of Stage 3 is also random but with a minimum value of 60 a as available TMA aerial imagery

shows ice at our site was in its present-day configuration in 1962 (Supplemental Fig. S3).

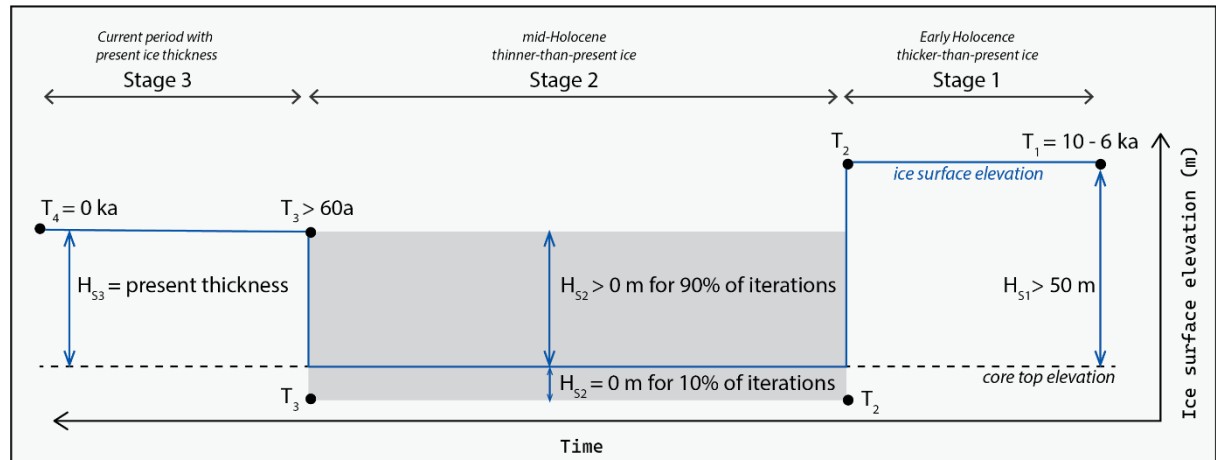

**Figure 4. Schematic of the forward model we use to predict downcore nuclide concentrations under varying ice thickness histories. $T_1$ is when the model begins, $T_2$ is the onset of hypothesised Holocene low stand, $T_3$ is end of low stand, and $T_4$ is the present day. $H_{s1}$,**
**$H_{s2, and}$ $H_{s3}$ refer to the thickness of the ice column during Stages 1, 2, and 3 (S1-3) respectively.**

For each stage ($S_x$) we calculate total mass depths in the core ($z_{sum} = z_{(i)} + z_{(c)}$) where $z_{(i)}$ is the mass depth (g cm$^{-2}$) at the bottom

of an ice column $H$ (cm) thick with a density $(\rho_{ice})$ of 0.917 g cm$^{-3}$:

$$z(i) = H(S_x) . \rho_{ice}$$

the mass depths in the core ($z_{(c)}$) are constant for each stage of the model and defined by:

$$z(c_x) = D(x) . \rho_{rock}$$

where $D(x)$ is the depth in the core (cm) and $\rho_{rock}$ is an assumed rock density (2.68 g cm$^{-3}$). The total mass depths ($z_{sum}$) are

used alongside the surface elevation of the ice sheet during the corresponding stage to calculate downcore nuclide production

rates during each stage of our model ($P_{S1}$, $P_{S2}$, $P_{S3}$ in atoms g$^{-1}$ yr$^{-1}$). Production rates are calculated as the sum of spallogenic

and muonogenic production as implemented by Balco et al. (2023) using the scaling method of Stone (2000).

The resulting nuclide concentration for each stage of our model ($N_{Sx}$) are calculated at each depth in our core as:



$$N_{Sx} = N_{Sx-1}.exp(-\lambda.T_{Sx}) + (\frac{P_{Sx}}{\lambda}.(1 - exp(-\lambda.T_{Sx})))$$

Where $N_{Sx-1}$ is the nuclide concentration at the end of the preceding stage (assumed to be zero at the start of the model), $\lambda$ is the decay constant of $^{14}C$ (1.2097 x $10^{-04}$ year$^{-1}$), $T_{Sx}$ is the duration of the stage ($S_x$) in years (e.g., $T_{S1} = T_1 - T_2$), and $P_{Sx}$ is the depth specific production rate for the relevant stage of the model (atoms g$^{-1}$ year$^{-1}$).

In all boreholes we encountered ice that was completely frozen to its bed. Considering low mean annual temperatures at our site (< -20°C; cf. Wang et al., 2023), basal melting under present-day ice thickness is glaciologically implausible making the occurrence of significant erosion during the most recent period of (thin) ice cover (Stage 3) unlikely. Similarly, while sub-aerial erosion does occur in Antarctica quantified rates are generally very low (< 1 mm ka$^{-1}$), including in quartzites (Marrero et al., 2018). Combined with observations of striations on polished bedrock surfaces above ice we consider it likely that sub-aerial erosion during any period of Holocene exposure (Stage 2) was negligible. Finally, any erosion during thick ice cover (Stage 1) is not relevant to our model as the short half-life of $^{14}C$ (5730 years) requires any pre-LGM inventory to have decayed regardless of whether erosion occurred. Consequently we assume zero erosion in all stages of our model.

## 4. Results

### 4.1 Luminescence depth profile (BH01)

On visual inspection the luminescence depth profile measured in BH01 (Fig. 5) hints at the presence of a potential bleaching front in the upper ~2.5 mm of the core. Specifically, slices 2 and 3 (depths 1.5 and 2.5 mm respectively) produce normalised $D_e$ values that are lower than eleven of the twelve slices from greater depths. However, notably the $D_e$ value of slice 9, from 8.5 mm depth, overlaps the upper bound of the 3.5 mm slice. Additionally, identification of a luminescence depth profile in BH01 is complicated by slice 1 (0.5 mm depth) which has a higher $D_e$ value than both slices immediately below. The measurement of higher luminescence and/or $D_e$ values for surface slices of depth profiles has previously been reported but it is currently unclear what causes this (e.g. Luo et al. 2018, 2022). An assessment of the mineralogy of our core slices using SEM-EDS analysis does not indicate any significant changes in mineralogy between slices (Supplemental Fig. S6). Similarly, downcore RGB values, measuring the rock opacity for each slice, do not display a pattern that obviously explains the variations in measured $D_e$ values (Supplemental Fig. S7).





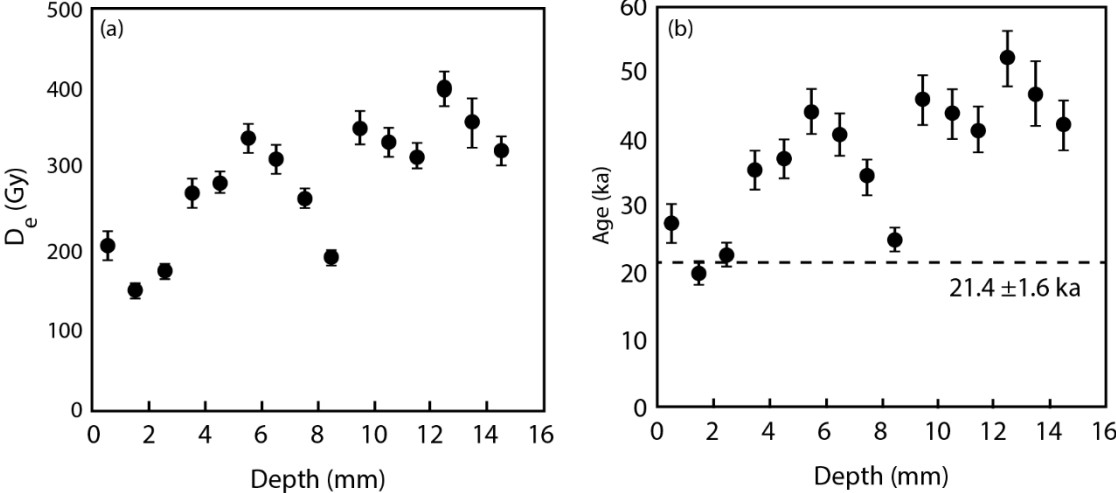

**Figure. 5. Luminescence depth plots for sample BH01 showing $D_e$ values (a) and apparent burial ages (b) measured using the OSL signal of quartz. Note that only Discs 2 and 3 (depths 1.5 and 2.5 mm, respectively) were used for age calculation.**

Although we cannot confidently identify a bleaching profile in sample BH01, at face value the potential shallow depth profile in BH01 (up to 3.5 mm depth) would be consistent with the dark opacity of the rock (e.g. Ou et al. 2018) which will have increased light attenuation with depth into the rock during any sunlight exposure, reducing the depth to which the luminescence signal would be reset. However, using the depth-specific environmental dose rate produces an apparent burial age of c. 21.4 ± 1.6 ka (Fig. 5b) for slices 2 and 3. Such an age would imply exposure to light during the LGM when the ice-sheet surface across the WSE was several hundred metres thicker than present and the grounding line was >300 km outboard of its present-day limits (Bentley et al., 2010; Hillenbrand et al., 2014; Hein et al., 2016; Nicholls et al., 2019; Johnson et al., 2019).

Although the sample passed dose-recovery experiments, it is possible that the luminescence signals of these rock slices were not dominated by quartz and thus were not suitable for measuring burial ages for a natural sample. Previous studies have reported dim luminescence signals measured for quartz from rock slices (e.g. Sohbati et al. 2011) similar to the signals reported here for sample BH01 (Supplemental Fig. S8); this is likely due to the lack of sensitisation of the signal prior to burial within a rock. The SEM-EDS measurements of sample BH01 (Supplemental Fig. S6) show that K is present in the rock slices (potentially related to the presence of K-feldspar) This would be consistent with measurement of high fading rates measured (9.2 ± 2.3 %/ per decade). It has previously been shown that feldspars emit detectable signals when stimulated with blue wavelengths and detected in UV (e.g. Thomsen et al. 2008). As such, the dim quartz signals mean give greater opportunity for minerals with less suitable luminescence properties (e.g. feldspar when stimulated and detected in blue and UV wavelengths, respectively) to dominate the signal used for dating. We suggest that this could have caused overestimation of the true burial age reported here for BH01.





245 **4.2 In situ $^{14}$C concentrations in bedrock cores**

The nine samples analysed for in situ $^{14}$C yielded total inventories of $4.5 \times 10^4$ - $9.1 \times 10^4$ atoms $^{14}$C (Table 2). The reported total inventories of $^{14}$C are ~3-6 times higher than the long-term blank value of $1.5 \times 10^4$ atoms $^{14}$C (Fig. 6). None of the samples overlap with the long-term blank value at 68% confidence and only one sample overlaps at 95% confidence. As argued by Balco et al. (2022), a key point is that if upper core samples contain inventories of $^{14}$C that are above background then, 250 given production systematics, $^{14}$C must be present in down core samples even if those samples have inventories that are close to blank levels. The uppermost samples from BH02 (5-10 cm), BH03 (Core top), and the single sample from BH06 are 2.9, 4.3, and 3.4 times above blank respectively, and do not overlap with the blank at 95% confidence. We thus conclude that in situ $^{14}$C is present in our cores. The generally observed downcore decrease in concentrations of $^{14}$C (atoms g$^{-1}$) in both BH02 and BH03 is consistent with production systematics of cosmogenic nuclides and further support this conclusion.

255

| Sample | Oz-Code | pmC | Δ pmC | C [µg] | Qtz Mass (g) | Total N$_{14}$ [at] | Δ Total N$_{14}$ [at] | N$_{14}$ [at g$^{-1}$] | Δ N$_{14}$ [at g$^{-1}$] |
|---|---|---|---|---|---|---|---|---|---|
| BH02 0-5* | 12602 | 7.09 | 0.29 | 10.92 | 1.752 | 45243 | 1854 | 16983 | 5494 |
| BH02 5-10 | 13403 | 5.78 | 0.78 | 15.85 | 3.853 | 53545 | 7206 | 9878 | 3084 |
| BH02 25-30 | 12604 | 2.97 | 0.25 | 27.68 | 4.643 | 48083 | 4123 | 7020 | 2220 |
| BH02 35-45 | 12605 | 4.58 | 0.23 | 25.44 | 4.815 | 68100 | 3499 | 10928 | 2093 |
| | | | | | | | | | |
| BH03 Core top | 12606 | 8.18 | 0.29 | 14.02 | 2.450 | 67088 | 2552 | 21016 | 3980 |
| BH03 0-5 | 12607 | 6.54 | 0.61 | 23.81 | 4.309 | 90965 | 8433 | 17515 | 2939 |
| BH03 5-10 | 12608 | 4.81 | 0.44 | 20.91 | 4.885 | 58771 | 5330 | 8861 | 2221 |
| BH03 15-20 | 12609 | 4.62 | 0.25 | 23.69 | 3.823 | 63865 | 3500 | 12654 | 2635 |
| | | | | | | | | | |
| BH06 Core top | 12610 | 4.41 | 0.29 | 20.37 | 3.605 | 52479 | 3511 | 10261 | 2796 |

\* 75-125 µm size fraction, all other samples utilised 125-500 µm

**Table 2. Results of in situ $^{14}$C analyses of samples from bedrock cores. Total inventories are not blank corrected; concentrations include global blank correction of 15500 ± 9500 atoms.**





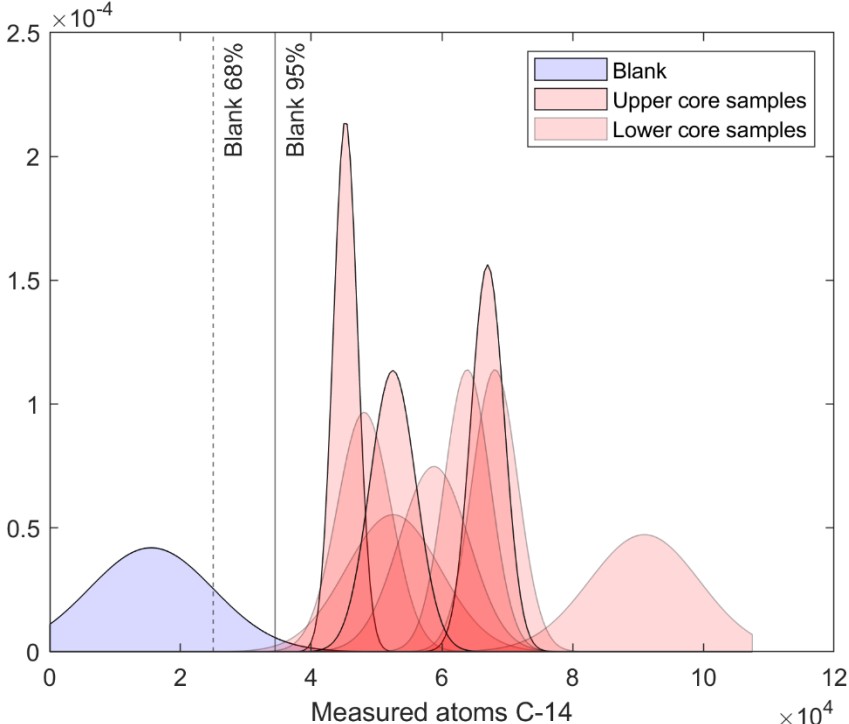

**Figure 6. Probability distribution functions showing total measured inventories (total number of atoms) of $^{14}$C in bedrock samples (red) and the global blank value of 15500 ± 9500 atoms (blue). Upper core samples are highlighted.**

## 5. Discussion

### 5.1 Was ice thinner-than-present during the Holocene?

In situ $^{14}$C can be produced at depths of tens of metres by deeply penetrating muons (cf. Lupker et al., 2015). Thus, it is possible that at least some of our measured concentrations could be produced without ice thinning beyond present if the duration of any quasi-constant ice cover was sufficiently long. To test this scenario we modelled expected downcore nuclide concentrations under constant (present-day) ice thickness for periods of 2 ka, 6 ka, and 8 ka. This represents a conservative time range as defined by existing constraints on ice-sheet history in the region (Bentley et al., 2010; Hein et al., 2016; Griemann et al., 2024). Notably, the predicted concentrations are significantly lower than the measured concentrations of the upper samples in both BH02 and BH03 (Fig. 7). In addition, the "straight" shape of the modelled profiles reflects the dominance of muonogenic production under ~25 – 29 m of ice (i.e. the present thickness at the sites of BH02/BH03). In contrast, the data from BH02 and BH03 define a production profile with a spallogenic contribution. These two observations support an inference that the nuclide concentrations in our bedrock cores result from production during a period when ice at our core sites was thinner-than-present allowing for increased production to occur.





Unlike the samples from BH02 and BH03, the [14]C concentration measured in the single sample from BH06 is consistent with, but does not mandate, production during prolonged (8 – 6 ka) burial by ice of present-day thickness which at this site is 9.5 m. However, given their respective locations, if ice thinning exposed the BH02/03 core sites then the BH06 core site must also have been ice free. Similarly, the OSL results presented here do not conclusively evidence past exposure of the BH01 core

surface to sunlight. However as with BH06, which is at a comparable depth, if ice thinning occurred at the sites of BH02 and BH03 then ice thinning must also have occurred at the site of BH01. We cannot exclude the possibility that both BH01 and BH06 were shielded by sediment and/or snow during any Holocene low stand, such a scenario could both shield BH06 from incoming cosmic rays, lowering the production rate, and prevent exposure of BH01 to sunlight. However, without observations such a scenario remains speculative. Importantly, we emphasise that the nuclide concentrations in both BH02 and BH03 show

an unambiguous signal of increased production above that possible due to muonogenic production under present-day ice thicknesses thus requiring ice to be thinner at our core sites at some time during the Holocene.

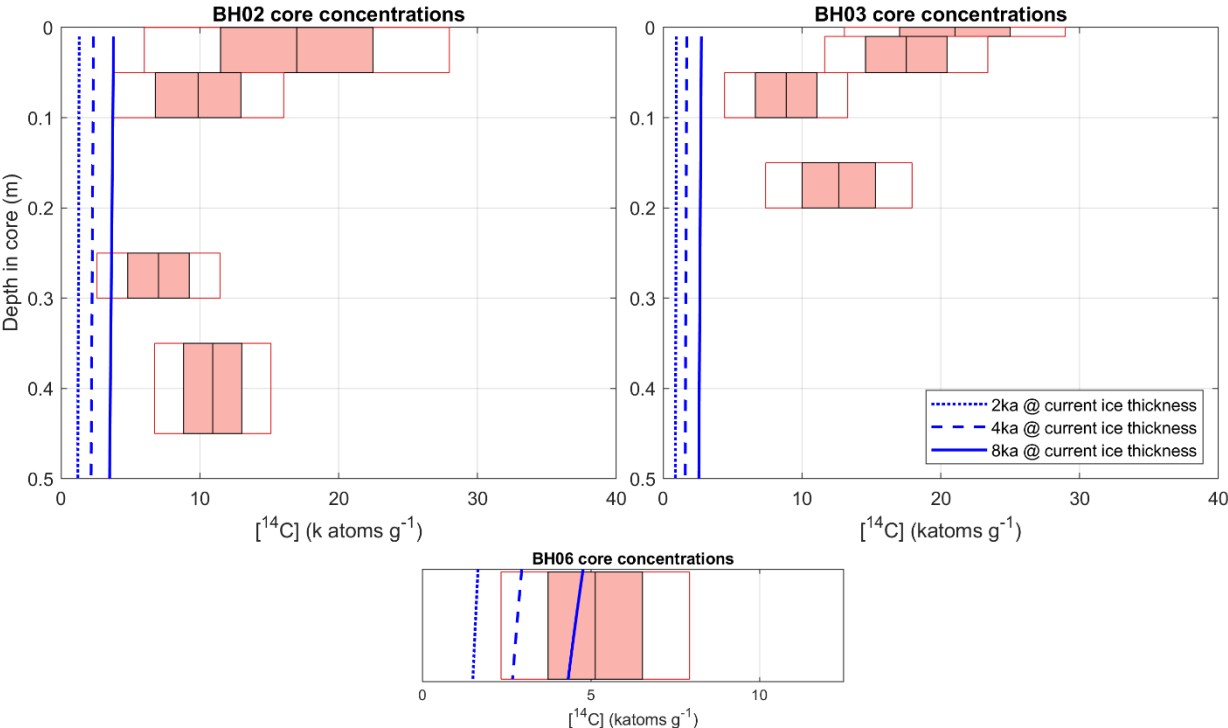

**Figure 7. Downcore measured nuclide concentrations for and predicted depth profiles under invariant ice cover for durations of 2 ka, 4 ka, and 8 ka for samples from BH02 (a), BH03 (b) and the single sample from BH06 (b). Solid box and outline denote 68% and**

**95% uncertainties on measured nuclide concentrations respectively.**





## 5.2 Ice surface histories inferred from forward model of nuclide concentrations

Our simple forward model of nuclide accumulation allows us to predict downcore nuclide concentrations in our bedrock cores under various synthetic ice histories (Fig. 8). Regardless of the relative contributions of different production pathways in situ $^{14}$C production rates decrease with depth (cf. Gosse and Philips, 2001). Consequently, as all depths in any individual core have

inherently experienced the same exposure-burial history the actual (i.e. not measured) concentrations must decrease with depth. This pattern in broadly observed in both BH03 and BH02. The predicted concentrations are compared to the measured concentrations to identify fitting model iterations. Fit is assessed simply using root mean squared error (RMSE) and we use the 5% of model iterations with the lowest RMSE in further discussion.

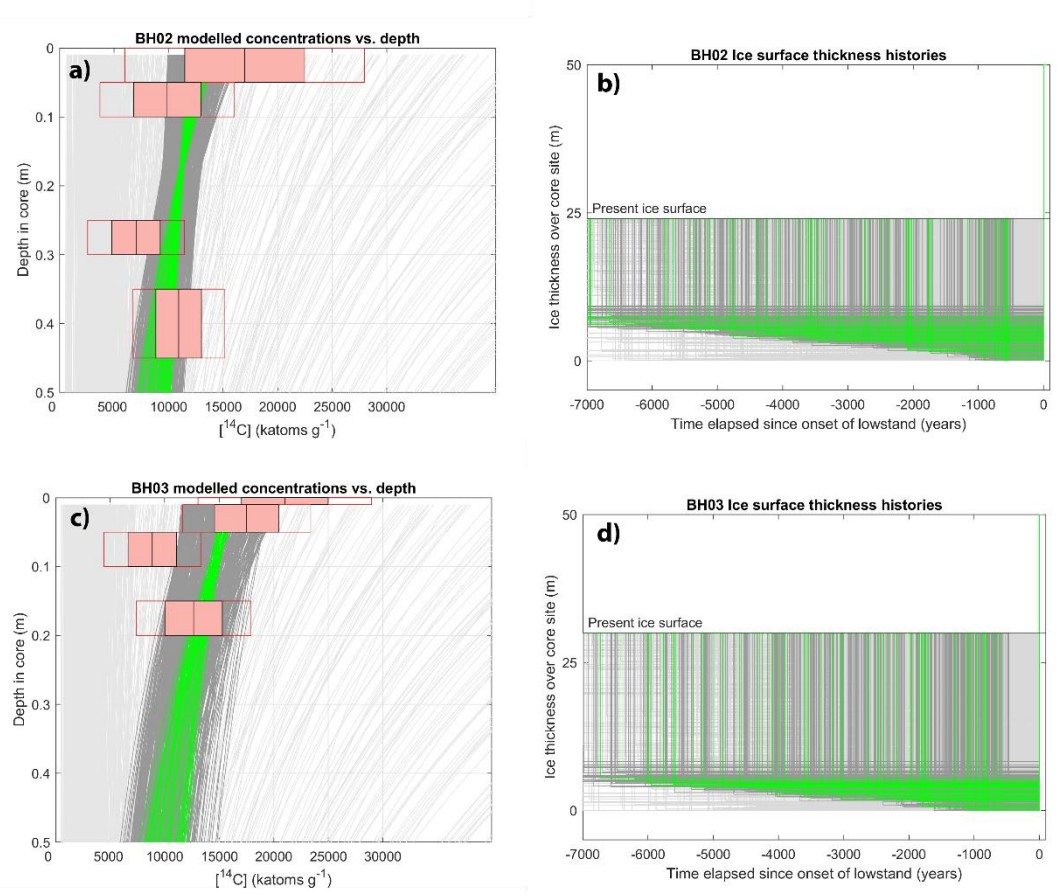

**Figure. 8. a) Predicted nuclide concentrations plotted against depth in core BH02. Measured concentrations shown as in Fig. 7.**
**Concentrations predicted by the best (10$^{th}$ percentile) of fitting iterations shown in green, top 5% of all iterations shown in dark grey, random selection (n = 1000) of all iterations shown in light grey. b) Three stage ice histories used to produce modelled concentrations, colours correspond to the same sub-sets of iterations as panel (a). The histories, which all have different starting times in the model, have been aligned to the onset of the low stand for visualisation. c) and d) are the same figures but for BH03.**





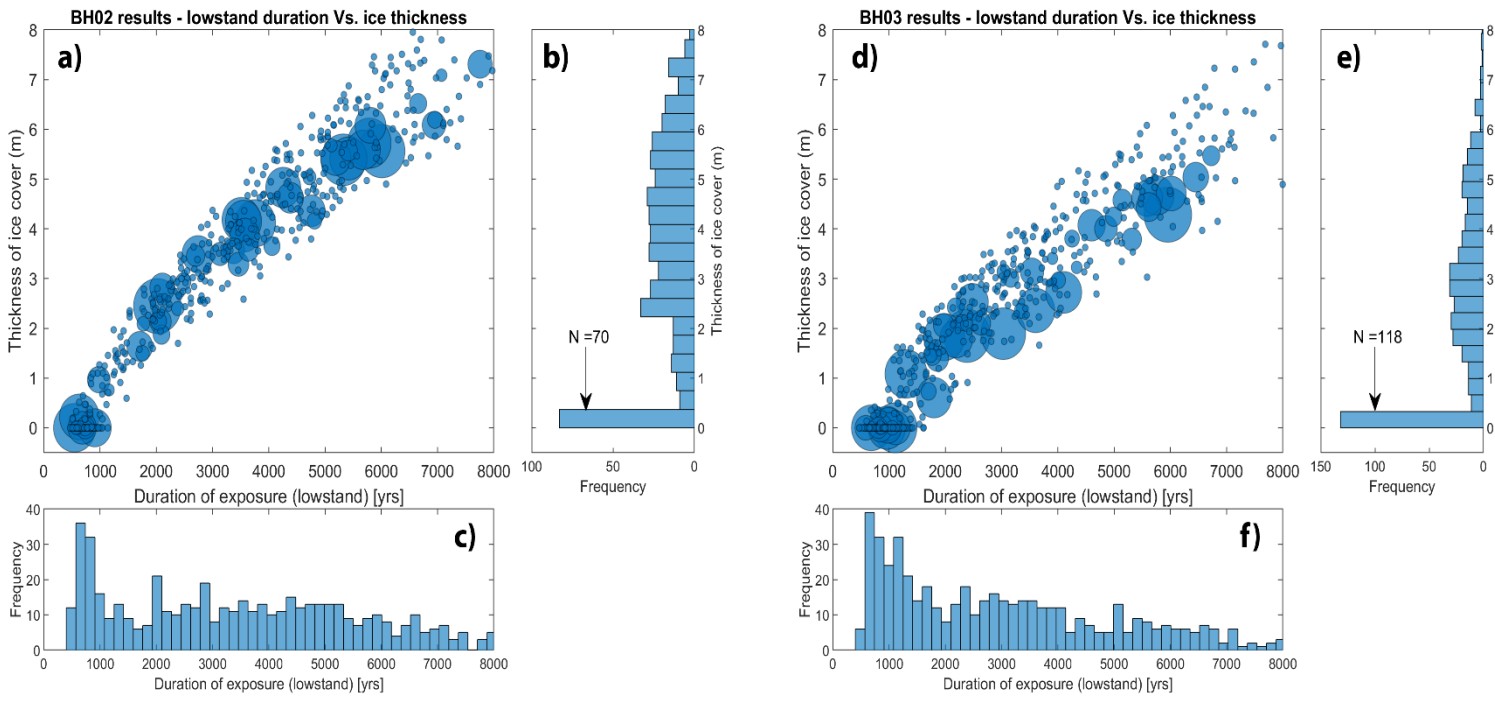

**Figure 9. a), d) Low stand duration plotted against thickness of residual ice cover for fitting model results from BH02 (a) and BH03 (b). The 10th percentile of best fitting scenarios has symbol sizes scaled in order of goodness-of-fit showing that scenarios of thin ice better fit the measured concentrations in BH03. (b, e) and (c, f) are the respective histograms for the two variables, again showing that shorter, 'none-or-thin' ice histories are preferred.**



The 5% of model iterations for both BH02 and BH03 suggest broadly comparable ice histories between our two core sites, as would be expected given their proximity. While nuclide profiles observed in both cores are consistent with a Holocene low stand during which our drill site was completely ice free (Fig. 9) our forward model, which is under constrained, also allows for scenarios with a thin residual ice cover, but relatively longer low stand durations. Put simply, if production rates in the

cores are reduced due to thin residual ice cover, then a longer time is required to accumulate the concentrations subsequently measured. Considering the full population of accepted scenarios the range of possible low stand durations is 500 – >8000 years with the upper limit of this range requiring residual ice thicknesses of ~8 m (Fig. 9). The range of possible solutions can be narrowed further by considering the total time available for thinning and re-thickening as constrained by the most proximal cosmogenic nuclide data from the Heritage Range. This data suggests ice thinned to near its present-day thickness by 4 ka (cf.

Bentley et al., 2010; Hein et a., 2016) which would limit the range of accepted solutions to encompass low stand durations of 500 – 3500 years and maximum residual ice thicknesses of ~4-6 m (Fig. 10). Notably, the 10th percentile of all best fitting



scenarios, shown by larger symbol sizes in Fig. 9, slightly favours histories where ice thinning either completely uncovered our sites or any remaining ice cover was thin (< ~3 m), particularly for BH03. These 'none-or-thin' ice histories correspond to Holocene low stand durations of c.500 – 4000 years, consistent with the potential timeframes as constrained by existing

exposure age data.

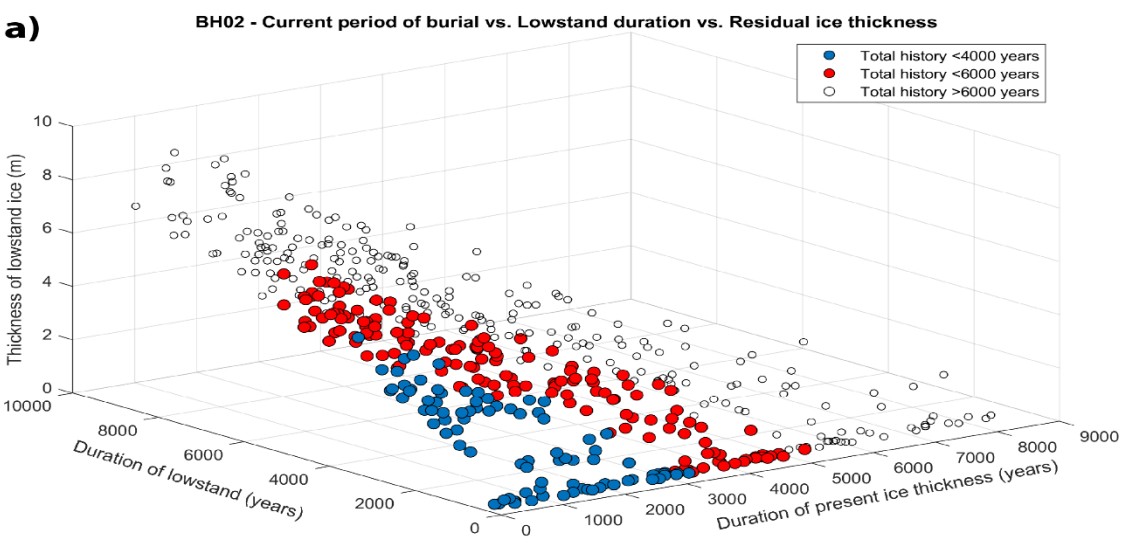

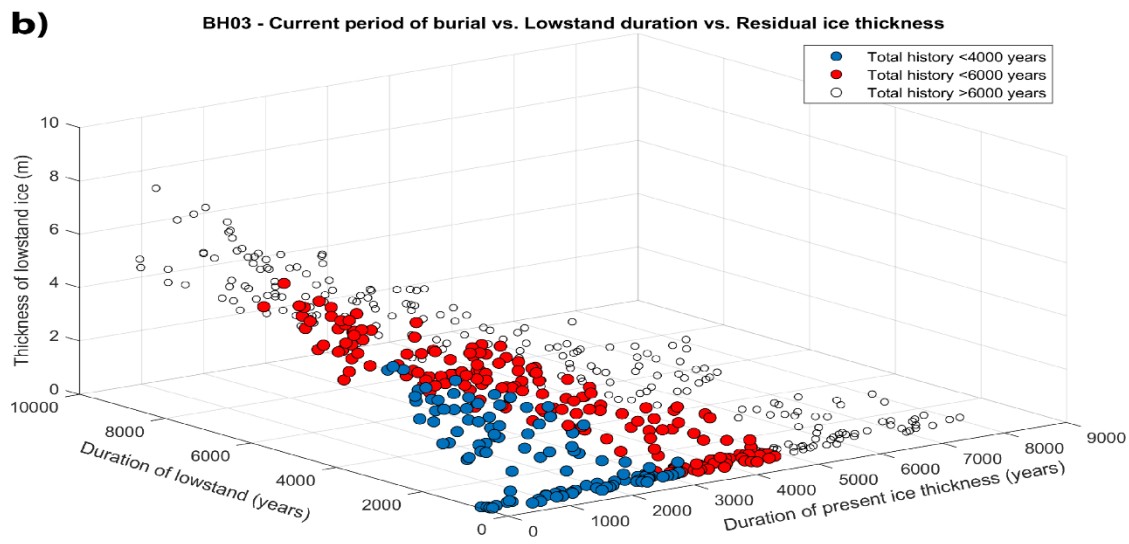

**Figure 10. XYZ plot of durations of low stand (Stage 2) and period of present ice thickness (Stage 3) against residual ice thickness during Stage 2 for cores BH02 (a) and BH03 (b). Nearest exposure age data imply ice surface reached modern elevations at c. 6-4 ka thus total duration of Stage 2 and Stage 3 would be shorter than this. Such constraints narrow the range of possible low stand**
**durations and residual ice thicknesses as shown for periods of 4000 years (blue markers) and 6000 years (red markers). Remaining accepted scenarios shown with white markers.**



## 5.3 Wider implications for Holocene ice-sheet history in the Weddell Sea Embayment

In summary, the data requires thinner-than-present ice at our core sites and is best consistent with a period of total, or near total, exposure. Such a scenario would constrain minimum, point specific, ice-surface lowering of ~24 m at the site of BH02 and ~29.5 m at BH03. The question then arises whether this represents a broader lowering of the regional ice surface (i.e. due to dynamic thinning associated with grounding-line retreat) or a (very) localised change due to site-specific factors such as wind scoop development and/or locally enhanced ablation in the BIA due to stronger katabatic winds.

Wind scoops are depressions eroded by vortexes formed when prevailing winds are obstructed (e.g., by protruding nunataks). Large examples can be eroded into glacier ice and can have floors several tens of metres below the adjacent ice surface (cf. Ackert et al., 2007). Wind scoops often form on the upwind side of obstructions, but their overall distribution reflects complex interactions of varying wind directions and local (meso-scale) topography. Presently, there is very limited wind scoop development around Plummer Nunatak, and none around the small outlier or Patch Rock. Given the lack of vertical relief around Patch Rock (i.e., < 2 m above ice) it is difficult to envisage development (and subsequent disappearance) of a wind scoop of sufficient size to expose our sample sites. Additionally, BH02 and BH03 are located on opposite sides of the subglacial ridge thus even if a wind scoop did develop it is unlikely that one would form on both sides of such a low relief outcrop.

Our core sites are located in an extensive Type 1 (cf. Bintanja, 1999) blue ice area (BIA) on the leeward side of the Enterprise Hills (Fig. 2b). Type 1 BIAs form where katabatic winds, enhanced by topography, focus sublimation on the lee side of nunataks (Bintanja, 1999). The first order control on the surface elevation of BIAs is regional thinning/thickening of the ice sheet over glacial timescales (cf. Woodward et al., 2022). Once a close approximation of the current ice-sheet configuration was reached (i.e. c. 6-4 ka) the predominant mechanism for localised lowering of the BIA surface elevation (in the absence of dynamic thinning) would be a situation where katabatic winds are enhanced leading to the rate of surface ice loss exceeding the rate of compensating ice flow. Katabatic wind strength is the product of continuous cooling over a sloping ice-sheet surface (Van den Broeke and Bintanja, 1995; Bintanja et al., 2014) thus wind strength increases with surface gradient. Minor elevation changes (< 100 m) at the WAIS divide during the Holocene (cf. Koutnik et al. 2016) would result in negligible changes in surface gradient over the >500 km between the WAIS divide and the Enterprise Hills. Additionally, modelled future changes in katabatic wind strength due to anthropogenic forcing are small (Bintanja et al., 2014). This implies that, in the absence of major ice-sheet elevation changes, variations in Holocene wind strength, when atmospheric forcing was of a similar (or smaller) magnitude, would also be minor.

Thus, while we cannot be definitive, we suggest that the most parsimonious explanation for a thinning, and subsequent re-thickening, of the ice surface at our sites is a dynamic change associated with fluctuations of the grounding line. If this is the





case, then the surface elevation change observed at our sites is indicative of broader retreat-readvance of the WAIS in the southern WSE. The timescales involved in thinning and re-thickening of the ice surface, as constrained by the cosmogenic nuclide data from BH03, are consistent with observations of late Holocene (<4 ka) flow reorganisation over Bungenstock and Korff Ice Rises (Siegert et al., 2013; Brisbourne et al., 2019). Similarly, the modelled duration of the low stand (500 – 3500 years) agrees with predictions from GIA modelling constrained by GPS observations in the southern WSE (Bradley et al.,

2015). Notably, Bradley et al. (2015) observe that their GIA model could potentially produce a better fit to GPS observations if there was "an increase in the spatial extent of the retreat-readvance". The ice-loading histories used as inputs to their GIA model focus retreat over the RSB, c. 200 km south of our site. Ice-sheet model simulations of past, and potential future retreat in the southern WSE also focus grounding line retreat in the RSB but generally not in the near vicinity of our site (Kingslake et al., 2018; Albrecht et al., 2020; Hill et al., 2021; Pittard et al., 2022). Our data therefore implies that Holocene retreat-

readvance of the WAIS in the southern WSE may have been more extensive than previously hypothesised. Constraining the overall extent of this change is a future research priority (Ross et al., 2025) and data from sites near the southernmost portion of the WSE, at the margin of the Foundation Ice Stream, are forthcoming (Small et al., 2024).

## 5.4 Comparison to evidence for retreat-readvance from other sectors of West Antarctica

While being mindful that different methods constrain different glaciological phenomena (i.e., grounding line retreat vs. ice-surface elevation change) it is worth comparing our data with evidence for Holocene retreat-readvance in other sectors of the WAIS. In the Ross Sea, radiocarbon dating of sub-glacial organic carbon recovered at and inboard of the Siple Coast grounding line indicates c. 250 km of grounding line retreat in the mid- Holocene c. 6.3 - 7.2 cal. kyr B.P, before readvance to the present-day position at an unconstrained time (Kingslake et al., 2018; Venturelli et al., 2020, 2023). In the Amundsen Sea cosmogenic

nuclide data from exposed nunataks, subglacial bedrock, and a sub-aerial controlled moraine (cf. Evans, 2009) constrains a minimum low stand duration of 3000 years that occurred sometime after c.7.1 ka with readvance occurring at c.1.4 ka (Adams et al., 2022; Balco et al., 2023; Nichols et al., 2024).

Our new data from subglacial bedrock cores in the WSE is consistent with these previous studies although we cannot constrain

the timing of ice-thinning directly. Indirectly, the nearest available above-ice cosmogenic nuclide data suggests that the ice surface initially reached it present elevation at c. 6.0 – 4.0 ka (cf. Bentley et al., 2010; Hein et al., 2016). If this was the case then excess thinning in the WSE must post-date this time, broadly in line with the timings in both the Ross Sea and Amundsen Sea sectors. The most likely duration of the low stand in the WSE (500 – 2500 years) is somewhat shorter than in the Amundsen Sea (>3000 years) although we note that our sites are located at higher elevations (~500 m asl Vs ~80 m asl) and further from

the present-day grounding line (~30 km Vs. ~2 km) than the core sites of Balco et al. (2023). Thus, the difference in modelled low stand duration may reflect a real variation or simply a delay between grounding-line retreat-readvance and corresponding ice-surface elevation changes at our upstream core sites. Our data, therefore, does not preclude the possibility that there was



near synchronous Holocene retreat-readvance of the WAIS in three of its main drainage catchments. Similarly, our data (and the other data constraining WAIS retreat-readvance) are consistent with recent global mean sea level (GMSL) reconstructions based on GIA modelling that suggest excess ice loss from Antarctica contributed to a higher than present GMSL since 7 ka (Creel et al., 2024).

## 6. Conclusions

The measured concentrations of in situ [14]C in two of our subglacial bedrock cores, combined with the observed downcore decreases in these concentrations, provides diagnostic evidence for a thinner-than-present ice sheet at our core sites during the Holocene. Our concentrations and their depth profiles cannot be explained by an ice surface that was maintained at its present elevation following the cessation of post-LGM thinning. A forward model of nuclide concentrations produced under synthetic ice histories constrains the duration of the ice surface low stand and thickness of any residual ice cover. Low stand durations of c.500 – 9000 years, and residual ice thicknesses of up to 8 m can explain our data however the best fit is provided by a low stand lasting 500 – 4000 years and ice no thicker than ~3 m. While we cannot exclude the possibility that ice surface change at our site is due to local factors, the lack of an obvious mechanism for such localised change, alongside the wider evidence for major ice-sheet change in the WSE during the Holocene, leads us to suggest that the thinning, and subsequent re-thickening demonstrated by our data reflects dynamic thinning associated with retreat of the grounding line inboard of present-day limits in the southern WSE. The areal extent of this retreat is greater than predicted by existing ice-sheet models simulations. Direct geological evidence for Holocene retreat-readvance is now observed in the Ross Sea (Kingslake et al., 2018; Venturelli et al., 2020, 2023), Amundsen Sea (Balco et al., 2023; Nichols et al., 2024) and Weddell Sea (this study) sectors of the WAIS. Retreat-readvance of the WAIS can explain a higher than present GMSL in the mid-late Holocene but the relative contributions (i.e., future vulnerabilities) of the main sectors of the WAIS have yet to be fully determined.

Finally, modelling studies have suggested that following excess retreat inboard of present limits in the WSE, ongoing glacio-isostatic adjustment led to re-grounding of ice rises, increasing buttressing and promoting grounding-line advance on a reverse bed-slope (Kingslake et al., 2018; Pittard et al., 2022). This mechanism has been proposed as a negative feedback on potential future sea-level rise from Antarctica (Gomez et al., 2015; Kachuck et al., 2020). Our data provides a testable scenario of retreat-readvance, along with potential timeframes involved, which should be used to test numerical ice-sheet models that include representations of this mechanism and are used to make predictions of future sea-level rise from Antarctica.



**Code and data availability**

All data described in this paper along with the MATLAB code used to carry out forward modelling calculations in this paper are included here and in the Supplement.

**Author Contributions**

DS conceived the study and obtained funding. DS, TL and ST conducted fieldwork and sample collection. DF, MMR and RHF prepared samples to clean quartz. RHF carried out carbon extraction. AS and RHF produced the cosmogenic nuclide measurement. RKS performed luminescence measurements. GVB oversaw training DS on use of the Winkie Drill and provided extensive guidance on its modification. DS developed the model for data analysis. DS prepared the initial manuscript with input from RHF and RKS. All authors reviewed the final version of the manuscript.

**Competing interests**

The authors declare that they have no conflict of interest.

**Acknowledgements**

Fieldwork was supported through the British Antarctic Survey and particular gratitude is given to the pilots and crew of the BAS Air Unit. DS, TL and ST would also like to acknowledge our BAS field guide, Sam Hunt for his extensive efforts in
facilitating the fieldwork. Antarctic Logistics and Expeditions are also thanked for logistical support. This work would not have been possible, or successful, without extensive support and onsite training for which DS thanks the U.S. Ice Drilling Program for support activities through NSF Cooperative Agreement No. 1836328. DS would like to thank Ian Garrett for his help with electrical components during fieldwork preparations. We acknowledge the financial support from the Australian Government for the Centre for Accelerator Science at ANSTO through the National Collaborative Research Infrastructure
Strategy (NCRIS). Thank you to Archana A. Joshi for preparing small carbon standards for the AMS run. We also thank Valerie Olive for making ICP-MS measurements.

**Financial support**

DS is supported by Natural Environmental Research Council Independent Research Fellowship NE/T011963/1. The GPR
equipment was provided through Natural Environmental Research Council Geophysical Equipment Facility loan 1144 to DS.





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
