# Peer review of "A thinner-than-present West Antarctic Ice Sheet in the southern Weddell Sea Embayment during the Holocene."

_EGUsphere, 2025_

## Author Comment (AC1)

**Reply to Reviewer 1 – Greg Balco**

***Please find our responses in bold italics below.***

Summary: This paper is very much of value because the relatively newly available method of subglacial bedrock exposure dating is really the only available means of obtaining data on what the Antarctic ice sheets looked like during past periods of warm climate when they were smaller. As we live in a warm period, understanding how and when ice sheets become smaller than they are now is, obviously, of critical importance for sea-level prediction. This paper is an important contribution in this area. Its main conclusion, that cosmogenic carbon-14 measurements on subglacial bedrock require a period of thinner ice in the middle Holocene, appears to be correct as far as I can tell. Overall this is a very good paper. However, the paper needs some attention in several technical areas having to do with the C-14 and luminescence measurements before I think it'll be suitable for publication.

I'm checking 'major revision' because I think publication requires (i) review by a luminescence specialist, and (ii) expanded reporting of process blank data for the C-14 measurements. See below for details.

***We thank the reviewer, Greg Balco, for the supportive and constructive comments and hope that we can address the points raised to satisfaction.***

Details:

The main conclusion of this paper is that the C-14 measurements on subglacial bedrock require a period of thinner ice in the mid-Holocene. As this is consistent with other indirect geological evidence that somewhere between permits and suggests a Holocene lowstand (really we probably shouldn't use 'lowstand' for this because of the confusion with sea level...maybe 'thinstand'?), this conclusion isn't unexpected, but it's extremely valuable to be able to show this with direct evidence and put some limits on the extent and timing. Starting from the reported blank-corrected C-14 concentrations, I repeated the ice-thickness-history fitting calculations with a similar random search optimization code and came to a similar conclusion as the authors, so, given the assumption that the reported C-14 concentrations are accurate, I agree with their conclusions.

There are some technical areas of the paper that need attention, as follows.

1. Luminescence measurements. Although I am not a specialist in luminescence measurements, I'm reasonably knowledgeable on the subject and I see several parts of this section of the paper that either don't make sense or seem incorrect. It should be noted, of course, that the luminescence data aren't involved in the main conclusion of the paper that there was a mid-Holocene thin period -- this conclusion rests completely on the C-14 data, so any issues with the luminescence measurements don't affect this main result. However, there seem to be some issues here.

First, the protocol used appears to be intended for quartz and is not, I think, able to differentiate between quartz and non-quartz signals. Thus, the later discussion that makes it appear likely that a lot of the signal is actually from feldspar would seem to have the implication that a lot of these results are pretty hard to interpret. Was a quartz protocol used simply because the rock is basically a quartzite in hand specimen scale, and it was not envisioned that significant feldspar would be present?

*Yes, the protocol is optimised for measuring luminescence of quartz but as mentioned in the text (cf. Thomsen et al., 2008) feldspars can also emit signal under this protocol. The answer to the second Q is simply; yes, pretty much. It was assumed quartz would be dominant following inspection of a hand specimen. Clearly, in hindsight this is regrettable, but it is what it is unfortunately. Hopefully by retaining a revision of this section (as the reviewer supports) we can help others avoid this mistake! There is presently very limited data available on rock luminescence dating of quartz so our paper would hopefully contribute to a growing understanding.*

*We suspect some of the issue here is that the conclusion that non-quartz minerals (i.e., feldspars) were important was added after (later) diagnostic experiments and thus after much of the section was originally drafted. It thus should have been re-written more fully than it was, we apologise for this. In short, we fully agree that the conclusion of a likely feldspar sourced contribution to the signal makes the results fundamentally hard to interpret. We had tried to convey this (as noted in the review), but the reviewer correctly points out that this is not as clear as it should be. We propose a simplification and rewriting of this section to emphasise this point more clearly (and avoid the interpretative preamble before we get to it) as outlined in the response below.*

Second, again, I am not really a specialist in this, but to me Figure 5 shows essentially no evidence of an exposure profile/bleaching front. I imagine this is a fairly clean quartzite, and the photos show a fairly light color, so it's probably a relatively translucent rock as rocks go, so I think we would expect a bleaching front > 1 cm deep, even for only a few years of exposure, no? The suggestion that there is a bleaching front present in the upper 3.5 mm therefore seems unlikely. Frankly, it would seem more feasible (or, I guess, less unfeasible) to argue that if there is a bleaching front, it is > 1 cm in thickness and manifested by the approximately linear increase in apparent age with depth (of course, that still leads to an impossible LGM apparent age). Note: there is some discussion of opacity in the paper and supplement, but there does not appear to be an actual measurement of the opacity that could be, e.g., compared with other rock types.

*All else being equal (i.e. zero erosion on the mm scale) then we would also expect a centennial scale period of exposure to result in a "relatively" deep bleaching front although without a calibration sample to quantify the attenuation coefficient and bleaching rate we can't ascribe a value to what would be expected. Previous work by two of the authors demonstrates that in relatively homogeneous sandstones bleaching fronts of ~*

*1mm and 4 mm are present after exposure periods of 0.01 and ~50 years respectively (Smedley et al., 2021). Thus, whether 3.5 mm is likely/unlikely is perhaps debatable but also probably moot given the inferred signal contamination by non-quartz minerals as the reviewer acknowledges in their previous comment. As with the previous reply we agree with the reviewer that fundamentally this data is hard to interpret and propose to rewrite this section to make that clearer and avoid interpretive discussion. Essentially, we propose to retain (and emphasise) the third paragraph of this section (lines 234 – 244).*

Third, the measured fading rate is extremely high, which would tend to indicate that we are looking at a saturation profile. Furthermore, the statement in line 243 that fading will result in a higher apparent burial age appears to be backwards -- fading should yield a younger apparent burial age, no?

*We do measure a high fading rate for this luminescence signal which, as the reviewer is correct about, could manifest as an age underestimation. However, as we state in the paper, we believe that it is a function of the luminescence signal used for measurements not being suitable for dating i.e. it is not a quartz signal, and it is not the K-feldspar signal stimulated by infra-red wavelengths that we would typically use for dating. The unsuitable luminescence properties could manifest in both age underestimation (because of the fading rate highlighted), but also in age overestimation as the signal may not deplete rapidly to sunlight exposure (for these minerals, at this wavelength). Essentially the data are not considered interpretable. Given the confusion here, we propose to re-write this paragraph to offer a clearer explanation of the results.*

In any case, I should point out again that I am not a luminescence specialist, but many parts of this discussion seemed incompletely thought out. It seems to me that really the only conclusion that you can get from the luminescence data is that it's not possible to reject the null hypothesis that both profiles are just saturated. This is basically the conclusion that the authors get to in the end -- the luminescence data provide no evidence for a completely ice-free surface in the Holocene -- but this section appears weak to me. I don't want to be discouraging of including these data: obviously we are all very new at collecting luminescence data from subglacial bedrock surfaces and I think it's extremely important to report all the available results, even if confusing, for the benefit of future work. However, I strongly suggest review of this section by a specialist in this field.

*We agree that all data and experience is valuable and thank the reviewer for their support in retention of this section once revised to satisfaction. The reviewer is correct, publishing this data is important to highlight that we may not be able to get reliable data from a quartzite by just using our established luminescence dating protocols for quartz, especially for samples with very limited material availability where measurements cannot be repeated.*

2. C-14 measurement. Basically, the review process for the Balco et al. 2023 paper that is obviously very familiar to these authors included a painfully long correspondence about exactly how to deal with analytical blank corrections for C-14 measurements, that eventually led to a large amount of reporting of process blank data for the Tulane C-14 laboratory where the measurements were made. The present authors have clearly absorbed some of this review correspondence (e.g., Figure 6), which is good. However, the authors have not provided the full set of process blank measurements as were eventually included in the Balco paper, which leaves the reader wondering a lot about the blue curve in Figure 6. Although the sense is that this value, which is described as a 'global' or 'long-term' blank, is probably a digest of multiple process blanks measured in the UoW lab, these constituent blanks aren't reported and there's no info about what the 'global' value actually is (mean and SD? Weighed mean?). Just as in the Balco paper, really the whole point of this paper hinges on whether C-14 has been detected above background and how you correct for that background. More importantly, the credibility of all our efforts at subglacial bedrock exposure dating using C-14 completely relies on this being right. False-positive detections of ice thinning using C-14 measurements would be extremely damaging to not only the credibility of the method itself, but also the credibility of the entire enterprise of ice-sheet and sea-level change studies. We can't have false positives.

Thus, the point of all this is that, just as in the other paper, we need to see all the process blank data before I think this is acceptable for publication. This will allow readers to see whether it's appropriate to use the proposed long-term blank or something more complicated than a normal distribution. The needed data here include the UoW lab and AMS data on all the process blanks that were used to assemble the 'global' value cited here, the dates they were run, and the dates that the samples were measured.

*We thank both reviewers for their comments concerning the in-situ ¹⁴C measurements and the performance of the UOW/ANSTO extraction line. We also fully agree that avoiding false positives is imperative. In response, please see the attached figure which we propose to include in revised supplemental information (alongside a revised Figure 6 in the main text). This shows the blanks associated with the samples from this study, listed in measurement order. The mean blank value (+/- s.d) for the past six years is 1.55± 0.94×10⁴ atoms. The 15 blanks processed alongside the current core samples are indistinguishable from this long-term average. Given this long-term reproducibility, the lack of any obvious time-dependence on reported blank values, and the fact that none of our samples overlap with any blank measured during the relevant period we do not consider additional blank modelling to be neccessary. The long-term blank produced by the UOW/ANSTO extraction line is discussed in more detail in a forthcoming paper by Fülöp et al.*

[Figure]

*Figure A1. Total ¹⁴C nuclide inventories of core samples and process blanks in measurement order. The six-year average (± s.d.) of blanks produced at UOW/ANSTO is shown with blue shading. The box and whisker plot summarises the distribution of the 15 process blanks analysed along with the unknowns.*

3. Optimization calculations. This is technical, and as noted above doesn't affect the overall conclusion, but I think the discussion of model fitting needs some attention.

First, we see (line 295) that we are discussing the best 5% of the random search models by RMSE, but we don't get any information about whether they actually fit in the sense of whether the actual value of the RMSE has a low rejection probability. As it happens, having redone the calculations, I know that it is possible to find lots of thinning histories that do have acceptable

RMSE by the usual criteria (that is, they're close to 1), which is what you would expect from the fact that the fact that the downcore measurements basically overlap at quoted uncertainties.

Specifically, what I found here is that for BH02 there are lots of histories that can't be rejected at 95% confidence, but for BH03 there aren't very many - there are lots at 98% but few at 95%. But for both cores you can find thinning histories that "fit" in a chi-squared/RMSE type sense.

However, there's no information in this section of the paper about what the relationship is between the top 5% by RMSE and the ones that actually fit based on a RMSE rejection criterion.

Are the 5% a small subset of the ones that fit the data (probably true for BH02)? Or are there histories in the 5% that don't fit the data (probably true for BH03)? I strongly suggest revising this discussion to focus on the histories that fit the data based on a statistical criterion and not just on the top 5%. [Note: this wasn't done in the Balco et al. paper because you can't fit the Be-10 data at high confidence -- there is a systematic misfit at the core bottoms that probably has to do with processes not included in the model -- but that issue doesn't apply here because there are no Be-10 data.] Adjusting this discussion won't have a significant impact on any of the conclusions, but it'll be a lot clearer as regards whether the models actually fit the data. In fact, as noted, they do fit the data according to usual metrics, but you can't tell that from this discussion.

*We are happy to amend our approach to model fitting.*

*Hypothetically, if we arbitrarily exclude BH03 5-10 then there are 438 fitting iterations at 95%. Obviously, we can't arbitrarily exclude a sample to obtain more fitting models. That said, the profile (at least qualitatively) follows the broad principles of decreasing concentration with depth. Thus we suggest it contains more useful information on the relatively wide range of ice histories that are broadly consistent with its profile than we could meaningfully discuss from only 2 fitting iterations.*

*Given that it is probably best to be consistent between the cores, and if the reviewer agrees with our point above that the BH03 profile as a whole contains useful information, then perhaps the simplest way to use a statistical criterion as suggested would be to focus further discussion on model iterations that can't be rejected at 98% confidence. This results in 1459 fitting iterations for BH02 and 255 fitting iterations for BH03. We can include some text on alternative cut off criteria of 95% confidence. We would amend subsequent discussion (and figures) which links to Point 4 below. Having done the preliminary calculations ahead of preparing this reply we would emphasise that, as stated by the reviewer these changes would not affect the overall conclusion of the paper.*

4. I think the statement in line 321-ish that the fitting calculations slightly favor the thinner ice/longer time models has a good chance of just being a random consequence of measurement uncertainty. Basically what that is saying is that the optimizer favors models that have a larger downcore decrease in concentration (because exposure happened closer to the

surface) over models that have a smaller downcore decrease in concentration (because exposure happened deeper).

What I found in redoing the calculations (with a piecewise-linear thickness change history as in Balco et al., not a piecewise-constant history as used in this paper) is that this effect is only present for BH03. This basically agrees with Fig. 6 in the present paper. I think this has a good chance of just being random, because the top 3 concentration measurements in BH03, although they are consistent with a surface spallation profile within their uncertainties, randomly happen to decrease faster than possible with a spallation profile. The optimizer is trying to fit this, even though it can't because the model can't get any steeper than a spallation profile, by making the ice thickness as thin as possible. Regardless, random or not, this effect does appear to be present in BH03. On the other hand, it's not present for BH02 -- the BH02 data do not favor the thinner/shorter solutions (or the opposite). So, this conclusion isn't super well supported. This is not to say that it isn't true that the lowstand ice thickness was on the thin side, just that it's tough to really prove it conclusively with these data. In the Balco et al. paper much of the leverage on this comes from the more precisely measured Be-10 data, and, of course, there are more C-14 data, so there is a bit more constraint on the slope.

Anyway, the conclusion, that the thin/short solutions might be favored, correctly isn't really strongly emphasized in the section of the text near line 321, so I think that part is fine. However, this needs some clarification in line 335-ish -- I am not sure what 'total exposure' means, but it would appear that the luminescence data preclude the rock surface actually being totally ice-free, so 'total' is misleading. It seems like really the most simple and specific way to state the constraint is that thinning has to be more than about 25 meters, period. And, also, it is worth noting that you could have more thinning -- you could easily have a situation, as is common now, where the rock surface is covered with permanent snow or thin ice (look at the big wind tail on Plummer Nunatak) but the outcrop as a whole is well above the regional ice surface.

*We are happy to amend the discussion around the points raised above. We agree that only BH03 shows a "clear" spallation profile that could be considered indicative of zero residual ice cover during the lowstand. Potentially, some of the random variation (i.e. concentrations decreasing faster than possible with a spallation profile or increasing slightly at depth) is due to variability in blank correction or the presence of mineral inclusions but this is wholly speculative.*

*We suggest (in line with the reviewer) that the simplest way of summarising the thickness change is the data indicates thinning of at least c.20 m but does not exclude the possibility that it was more than this (at least regionally).*

5. There's a lot of 'while we cannot exclude' and 'while being mindful that...' towards the end of the paper, which is fine, and correct because at a site like this the real relationship between dynamic thinning, blue-ice ablation, and the ice thickness is mostly unknown, but really the important point here is not that the data in this paper are ambiguous with respect to grounding line position, but instead that this type of data can only ever be collected in certain places

where there is rock to drill into. We can't ever go to the exactly perfect place that has the perfectly unambiguous relation to grounding line position, because that (probably) doesn't exist. So I would be clear, as the authors have, that nothing is perfectly unambiguous because nothing is in exactly the right place, but also point out that this is something that needs to be figured out! If we are serious about understanding sea level impacts of ice sheet change during past warm periods, glaciologists have to get to work on making the relationship between this type of data and grounding line positions less ambiguous. So it's fine to have all the caveats in the conclusions of this paper, but it is not good to give the sense that the authors are apologizing for the fact that they collected data from the best possible place closest to the grounding line that actually exists. Don't apologize! I would also point out that there is also an important challenge here -- the sites where we can collect this type of data are imposed by geology/topography, but we need to be able to quantitatively use this type of data, and even though that's not in scope for this paper, people need to get to work on figuring this out.

*We completely agree with the reviewers point that data of this type will (probably) never be from the perfect location. Indeed, in light of our experiences even choosing a place to drill includes some degree of faith that ground conditions will be amenable to drilling/rock recovery. You are always, to some extent, drilling blind. We also agree that we need to better understand what this data means for the magnitude of past GL retreat (linking to comment of reviewer 2). The point(s) raised here are strong and we are happy to include some brief reflective text and of course include the citation to this review.*

Minor items:

What is a 'sub-aerial controlled moraine' (line 385)?

*The opposite of the mythical subglacial controlled moraine? Suspect we have included an unnecessary extra adjective in this context!*

**Proposed list of changes**

- **Simplification of discussion of luminescence results to emphasise more clearly that the data is not to be interpreted.**
- **Include addition figures showing individual blanks measured alongside samples.**
- **Amend fitting approach to use statistical criterion and adjust following discussion and figures accordingly.**
- **Include some brief additional (reflective?) discussion on experience/choice of drill site (in line with comments of reviewer 2).**

---

## Author Comment (AC2)

**Reply to Reviewer 2**

*Please find our responses in bold italics below.*

The paper entitled, "A thinner-than-present West Antarctic Ice Sheet in the southern Weddell Sea Embayment during the Holocene" by Small and colleagues presents new cosmogenic nuclide inventories from subglacial bedrock cores that elucidates Holocene ice sheet history in the Weddell Sea Sector. This work is very timely as grounding line retreat and re-advance in the Holocene has been found in both Amundsen (Balco et al., 2023; Nichols et al., 2024) and Ross (Venturelli et al., 2020, 2023; Kingslake et al., 2018) sectors of West Antarctica. Thus, the results presented in this paper are important because they prove the hypothesis of retreat and re-advance in the Weddell Sector originally posed by Bradley et al (2015) and demonstrates that this phenomenon occurred in **every** sector of West Antarctica. As a result, I believe this paper should be published with very minor revisions and I provide some minor comments and questions below that I hope will help to clarify what is already a very excellent contribution to the literature.

1. The feat of drilling and recovering 4 subglacial cores is really impressive. I found this to be a bit underplayed in the manuscript and suggest including a bit of context to highlight (a) how impressive/important this is and (b) how targeting blue-ice areas might be advantageous in future subglacial drilling areas given the results and successes herein. It could also be useful to add some context about the conditions at the ice-bed interface, given past challenges with sediment-laden basal ice in non-blue-ice-areas described in Braddock et al. (2025).

***We are happy to add some extra detail on our drilling experiences in the methods section. The clean transition at the interface was mentioned but we will expand. We also drilled in a blue-ice area in 23-24 (Pensacolas) and encountered (some) sediment-rich basal ice, speculatively perhaps geology is a key factor here with certain lithologies producing more debris rich basal ice. That said we agree that the experience of all drilling campaigns is useful for future efforts and will include some extra context.***

2. I realize that some of this information is in the supplement, but can you provide a bit more information about quartz preparation for in situ $^{14}$C samples? With the work of Nichols & Goehring (2019) showing that quartz isolation/preparation methods may impact the validity of the in situ $^{14}$C result, adding some specific details about how long successive leaches were carried out, if any other separation techniques were used (e.g., frothing or magnetic separation), etc. would be useful for the interpretation of these results into the future.

*We will add the extra information on leach durations etc. Samples were initially cleaned at SUERC. After delivery to UOW/ANSTO one was randomly selected for purity testing. Although a small difference was noted between the reported and measured purity, it remained within the expected analytical range.*

*We will also clarify that no froth flotation was undertaken however previous experience in preparation of samples for in situ 14C extraction suggests that differences in quartz purification protocols have minimal impact on in-situ $^{14}$C results until the final 1% HF/HNO$_3$ etch is applied. Consequently, froth flotation remains a reliable and preferred method for producing clean quartz aliquots. All samples in the present study were extracted using an updated and more rigorous protocol than that published in 2019 by Fülöp et al. For this reason, we report intercomparison data CRN (three samples using the same protocols as our unknowns) with standard deviation of 14%. Full details of the revised protocol will be presented in the forthcoming paper by Fülöp et al.*

3. In addition to quartz preparation: I do have some specific questions about the in situ $^{14}$C data:

- Can you provide some more information about the long-term blank value and how that is derived? With the methods paper being cited coming from ~6 years ago, I do think more information is needed here. Specifically, I would like to see the values of individual blanks analyzed during the measurement window of the core samples presented herein. This would help to not only understand the long-term blank value, but also blank variability during the measurement of these samples.

  *As with our reply to reviewer 1, we thank the reviewers for their comments concerning the in-situ $^{14}$C measurements and the performance of the UOW/ANSTO extraction line. In response, please see the attached figure which shows the blanks associated with the samples from this study, listed in measurement order. The mean blank value (+/- s.d) for the past six years is 1.5 5± 0.94×10$^4$ atoms. The 15 blanks processed alongside the current core samples are indistinguishable from this long-term average. Given this reproducibility, and the fact that none of our samples overlap with any blank measured during the relevant period we do not consider additional blank modelling to be necessary. The long-term blank produced*

*by the UOW/ANSTO extraction line is discussed in more detail in a forthcoming paper by Fulop et al.*

[Figure]

**Figure A1. Total $^{14}$C nuclide inventories of core samples and process blanks in measurement order. The six-year average (± s.d.) of blanks produced at UOW/ANSTO is shown with blue shading. The box and whisker plot summarises the distribution of the 15 process blanks analysed along with the unknowns.**

- I am curious about the very low CRONUS-A concentration (5.85 x $10^5$ $^{14}$C atoms g$^{-1}$) presented in the supplement. The consensus value for CRONUS-A is 6.93 ± 0.44 ×$10^5$ $^{14}$C atoms g$^{-1}$ (Jull et al., 2015), which has been maintained in the literature as new laboratories come online (Lupker et al., 2019; Fulop et al., 2019; Lamp et al., 2019; Lifton et al., 2023; with the exception of similarly low values from Goehring et al (2019)). The authors state that an in vacuo cleaning step at 600°C was used to remove meteoric $^{14}$C—how long was this step carried out? Lifton et al (2023) showed that holding a sample at 600°C for >1 hr has the potential to remove higher temperature/in situ $^{14}$C, so it is possible that this step is removing some of the in situ component here too. These methodological details are important as they have implications for the interpretation of ice history from the core samples, because concentrations presented might not reflect the full period of exposure, but rather a concentration accumulated during exposure minus the removal during the 600°C step. If this explanation can not be used to justify why the CRONUS-A value is so much lower than the consensus value, I do think some justification and details on the value

presented are needed (How many measurements of CA were done? Was there any variability? Why doesn't the value presented here match previous values from this lab (e.g., Fulop et al., 2019).

*We agree that the CRA value quoted here will be of interest to the wider community however in the first instance we would note that the intended purpose of quoting this value was to emphasise that we considered it unlikely that we were systematically overestimating inventories (cf. Reviewer 1's comment on false positives). The value is derived from a significant body of work (representing significant effort as I am sure the reviewer will appreciate) and to go into substantial depth on, the overall distribution of values as well as potential explanations for the observed values is beyond the scope of this reply (or paper). That said it is important to point out some key methodological differences between the approaches described by Lifton et al. and by Fülöp et al. which make comparisons complex. These differences fall into three main categories: flux use, oxygen availability, and dilution strategy.*

1. ***Flux versus no flux.***
   *Lifton et al. employ lithium metaborate flux, which melts at 845 °C and reduces the effective melting temperature of quartz to ~1200 °C, allowing complete melting. In contrast, Fülöp et al. heat quartz to ~1650 °C without flux, inducing a phase transformation rather than melting the mineral. It is possible that quartz softening in the Lifton et al. system begins at lower temperatures (~600 °C), particularly in the presence of additional oxygen, but this has not been directly demonstrated.*

2. ***Role of oxygen in fluid-inclusion degassing.***
   *In the Fülöp et al. method, samples are heated in vacuo, without oxygen addition. Quartz does not release the major fluid-inclusion component until temperatures exceed the α–β transition at 573 °C, consistent with results presented in Fülöp et al. (Goldschmidt 2019; Radiocarbon 2022) and there is no loss of in-situ C-14 signal. By contrast, the introduction of oxygen may enhance degassing and potentially alter the extraction pathway, although the presence and magnitude of this effect remains uncertain.*

3. ***Blank dilution and carrier strategy.***
   *Lifton et al. (and Goehring et al.) routinely dilute samples and blanks by factors of 10–40. Under these conditions, even small sources of contamination can become masked by the blank gas, making it difficult to distinguish laboratory contributions derived [14]C. Their flux also contributes to the procedural blank, as shown in Lifton et al. (2023). Furthermore, no synthetic quartz is used to replicate the matrix behaviour.*
   *In the Fülöp et al. approach, synthetic quartz is processed through the same extraction pathway, and solely from the extracted quartz the gas remains within the uncertainty of the pressure transducer unless a solid diluent is introduced before extraction-not after. This enables more direct monitoring of the blank contribution.*

*Regarding the CRA reference material. Supplementary data in Balco et al. (2023) show that the Tulane laboratory reported an average CRA value of ~5.88 x 10⁵atoms which is statistically indistinguishable from the UOW/ANSTO value. For other laboratories, meaningful comparison remains difficult due to methodological differences such as those highlighted above, and any discrepancies should be properly evaluated once the consensus value reported by Jull et al. (2015) is systematically revisited by the community. We support such an effort, however we don't consider this paper the most appropriate place to tackle such an in-depth and technical question. We can add some text either in supplementary info or the main body highlighting the issue but pointing out, importantly that it does not change the key finding of this paper, namely that ice must have been thinner at some point during the Holocene. Finally, a forthcoming paper (Fülöp et al.) will discuss CRA values from the UOW/ANSTO lab in significantly more detail than is possible in this paper and we are reticent to pre-empt this paper and its peer review.*

- The slightly higher value at depth in BH02 and BH03 is a pretty cool observation, and it's interesting that the increase in concentration occurs at different depths. I think this is a bit overlooked in the text, and some context about how much muogenic production would be needed to produce the concentrations observed would be of value to the in situ ¹⁴C community.

*Prolonged muon production would, presumably, result in broadly similar concentrations in lower core samples. Our sense is that while we can be confident that in situ 14C is present in the deeper BH-02 and BH-03 samples (cf. blank values) their measured 14C concentrations are low enough to be influenced by variability within the blank value. That is perhaps the correction is "too large/small" on some samples and the produces an apparent increase in concentration at depth. We are thus reluctant to speculate on the apparent increase observed in the deeper BH-02 and BH-03 samples.*

4. The authors invoke dynamic thinning to explain that grounding line retreat would be expected alongside the magnitude of thinning observed from this work. This argument could really be strengthened by the addition of some simple calculations to demonstrate the likely magnitude of grounding line retreat associated with the thinning observed at the site. Such a constraint would be a valuable addition for constraining ice sheet models, and provide constraints on future sub-ice drilling efforts for grounding line retreat signals (e.g., Venturelli et al., 2020).

*This certainly crossed our minds however we decided against this for several reasons. Firstly, models show that grounding line retreat is focussed over the Robin Subglacial Basin which lies some distance from our drill site. We suspect thinning at our site is most likely linked with GL retreat in Hercules Inlet. Thus a quantification of the magnitude would be very specific to this small section of the wider GL and probably not reflect the (potential) major GL retreat (>300 km in some models) across the RSB. Secondly, it was not*

*obvious to us how to extrapolate upstream thinning to GL retreat in a relatively slow flowing area. Konrad et al. (2018) provide a value to link metres of thinning to metres of grounding line retreat (110 metres of retreat for 1 metre of thinning) in fast flowing ice streams (800 m $a^{-1}$) but it is unlikely that this value is appropriate for areas for slower flowing areas such as between our site and the grounding line (<20 m $a^{-1}$). We think this comment links to a point made by reviewer 1, specifically that we (the community) probably need to figure out how to better link the sort of data presented here to the former grounding line positions. Instead of speculating about the magnitude of retreat (for reasons above) we propose to make this point with reference to these reviews.*

Again, I believe this paper is a valuable contribution to the literature and the suggestions and requests above should be viewed only as minor suggestions. I really enjoyed reading the submitted manuscript, and I commend the authors on the excellent work presented herein.

*Thank you, we appreciate the review and suggestions which will improve the paper.*

**Proposed list of changes**

- **Add extra text on drilling experiences.**
- **Add extra detail on sample preparation.**
- **Include addition figures showing individual blanks measured alongside samples.**
- **Include comment on CRONUS A values (our choice would be to do this in the supplemental).**
- **Include additional point about linking data to magnitude of grounding line retreat.**